**www.cambridge.org/gmh**

## Research Article

mental health; global mental health; health disparities; healthcare workers; low-income countries

**Corresponding author:**
Youri Encelotti Louis;
Email: yourilouis@equalhealth.org

# Anxiety and depression screening reveals the importance of advancing mental health support for Haitian healthcare providers

Youri Encelotti Louis[1] , Brooke Betson[2] and Samy Auguste[3]

[1]EqualHealth, Montreal, QC, Canada; [2]University of St. Thomas, St. Paul, MN, USA and [3]La Paix University Hospital, Port-au-Prince, Haiti

## Abstract

Despite the recent momentum of mental health advocacy and resource allocation in several nations worldwide, the same progress is yet to be experienced in Haiti and other countries in the global south. In addition to the ongoing humanitarian crisis that continues to pre-dispose the people of Haiti to a variety of health conditions and mental illnesses, Haitian healthcare providers face further vulnerability to mental illness due to the high-stress nature of their work in a resource-limited environment. This study was conducted using a self-report questionnaire containing the Generalized Anxiety Disorder-7 and Patient Health Questionnaire-9 screening tools, distributed to Haitian healthcare providers nationwide. The results revealed that 84% of the 106 participants experienced mild to severe symptoms of depression, while 69% reported mild to severe symptoms of anxiety. This study also found that social determinants, including community violence, economic and social instability, and poverty, are among the most detrimental contributing factors to the mental health of Haitian healthcare providers. Despite the acute need for support, 76% of participants also reported having either no awareness or no access to mental health support. These findings serve as an urgent call for action to improve access to mental health resources for Haitian healthcare providers.

## Impact statement

The Caribbean nation of Haiti remains the poorest country in the Latin America and Caribbean region and has been engulfed in economic and social crises for centuries. As a result, its citizens are predisposed to a variety of medical conditions, including mental illness. This study sheds light on the mental health of healthcare providers in Haiti, who graciously provide their services amidst the ongoing turmoil. This study reveals that the effects of socioeconomic and political crises are among the factors with the most considerable impact on the mental health of the healthcare provider community. Despite the damaging effects of these factors, this study also reveals the substantial lack of mental health support and resource accessibility for healthcare providers across Haiti. Resource scarcity triggered by the ever-present effects of historical trauma and long-lasting exploitation of Haitian resources is the primary factor that impacts both the investment in clinical mental health services within the Haitian healthcare system and the support that it provides for its practitioners. The interaction between resource limitation and the cultural perception of mental illness often leads to the under-diagnosis of mental health disorders and an overall underestimation of the burden of mental illnesses in the healthcare provider community and beyond. This study reveals this widely unexplored issue to the general public and highlights the importance of making mental healthcare for health professionals available and accessible in clinics, hospitals and community centers with a collective and multidisciplinary approach that takes into account the cultural roots of the population.

## Introduction

In contrast to historical norms that acknowledged health as living an active lifestyle and being without substantial injury or illness, modern health institutions have transitioned to a more comprehensive understanding of health and well-being. The principle of health has been recognized as "complete physical, mental and social well-being and not merely the absence of disease or infirmity" by the World Health Organization (WHO; 2022, Constitution n.d.). An additional critical component of the revolutionized approach to health is the emphasis on mental health due to its ever-increasing burden on global communities. Likely as a result of international de-stigmatization efforts and mental health awareness, the WHO (2019b) has reported a 13% increase in worldwide mental health and substance use disorders since 2017. Also among the factors that contribute to the modern emphasis on mental health is the recognition of depression

serving as the "leading cause of disability worldwide" (WHO, 2017). In response to the ever-increasing severity of mental illness, several countries across the globe, including Canada, the United States and several European nations, have developed substantial momentum in their dedication to implementing systematic change and providing mental health resources in many other areas of the community.

In addition to systematic implementations, such as mental health policies and procedures, several useful resources have also been designed for clinical settings. Among these resources are the Patient Health Questionnaire-9 (PHQ-9) developed by Spitzer et al. (1999) and the Generalized Anxiety Disorder (GAD-7) questionnaire developed by Spitzer et al. (2006). These assessments are frequently used by healthcare providers as general screening tools for depressive and anxiety disorders in clinical settings. They are also used to monitor symptom severity in patients with a prior diagnosis of a depressive disorder (PHQ-9) and various anxiety disorders (GAD-7) (Kroenke et al., 2001; Spitzer et al., 2006). Both tools satisfy the *Diagnostic and Statistical Manual of Mental Disorders* (DSM) criteria and have been identified as accurate detection and monitoring tools for depression and anxiety disorders (Kroenke et al., 2010).

Despite the extensive research and the resources that have been allocated to combat the threat to global public health posed by mental illness, Pendse et al. (2019) suggest that providing adequate mental health support has yet to be prioritized in the global south. Wainberg et al. (2017) suggest that the shortage of mental health professionals, stigmatization and diminished clinical research capacity are among the most substantive factors inhibiting the development of mental health advocacy in these regions. Among the southern nations that have yet to identify mental health disorders as a significant threat to public health is Haiti. Considering Haiti's ongoing humanitarian crisis that is fueled by consistent political instability, natural disasters, violence and poverty (Overview, n.d.), its citizens are particularly vulnerable to various illnesses, including mental health conditions. After all, Haiti has faced a catastrophic magnitude 7.0 earthquake, a devastating cholera outbreak, multiple controversial federal elections, Hurricane Sandy, the assassination of President Jovenel Moise, years of paralyzing violent protests and the COVID-19 pandemic, all within the past 13 years (BBC, 2019). The extreme state of poverty that affects an astounding 86.7% of Haitian citizens, according to the (Overview, n.d.), is also paramount in the nation's susceptibility to illness, considering over 80% of people who are most severely impacted by mental disorders reside in low- and middle-income countries (LMICs) (Rathod et al., 2017).

Haiti's vulnerability to mental health conditions (Bolton et al., 2012) as well as its resource limitations is heightened by the ever-present effects of historical trauma and oppression that can be traced back to the massive exploitation of the nation's wealth that took place in the colonial era. Despite the pure heroism of the Haitian slaves who carried forth the "modern world's first successful slave revolution in 1791" and led what was then the island of Saint Domingue to gain independence from France in 1804, the nation's financial exploitation persisted (Porter et al., 2022). Despite their achievement of independence and the conquering of the colonial system that enslaved them, Haitians were left with debilitating and generational "independence debt" resulting from the reparations imposed by successors of French slaveholders. Haitian leaders were forced to succumb and pay crippling reparations to avoid another catastrophic war. France then ensured that Haiti would take out loans directly from its banks to pay the reparations, an act that historians would later define as the implementation of

Haiti's "double debt." As a result, Haitians were left not only with the sum of their "ransom" but also with massive loans that pushed the nation into a spiral of economic hardship that would extend into the present. Consequentially, Haiti was deprived and drained of all the necessary resources to invest in education, health and infrastructure. It has since been estimated that the nation has suffered ~115 billion dollars in losses (Porter et al., 2022). Following France's footsteps, the U.S. approach to the Haitian economy has been predominantly extractive. During the American occupation in 1915, salaries and expenses of the American officers were covered by Haiti's budget, while the national treasury was under complete U.S. control. In consideration of the continuous international manipulation that followed, Haiti has since been unable to operate without interference from foreign forces (Porter et al., 2022).

After the 2010 Haiti Earthquake, considerable efforts were made by Partners In Health and its sister organization, Zanmi Lasante, to support the government in developing a functional national mental health system (Raviola et al., 2020). Despite these efforts and the heightened national medical demand following the 2010 earthquake and recent cholera and COVID-19 outbreaks, Haiti's national expenditure on health remains critically below average for LMICs as their percent GDP spending was recorded at 3.22 compared with the average 5.61 across all other LMICs in 2020 (*Global Health Expenditure Database*, 2020; *World Bank Open Data*, 2020). Furthermore, the Haitian health system itself is broken, considering the quality and accessibility of healthcare services have been declining for years. For instance, there is a striking lack of healthcare facilities to meet the needs of the general population (Turner, 2014). Remote villages and communities are affected most severely by this scarcity due to the additional geographic barriers and the centralization of healthcare services in the capital, which further limit their access to resources (Grelotti, 2013). Additionally, the percentage of the national budget allocated for health dropped from 16.6% in 2004 to 3.9% in 2018 (Hashimoto et al., 2020). Of the limited national spending devoted to healthcare, even less resource allocation is devoted specifically to mental healthcare (WHO, n.d.). It was reported by Haiti's Ministry of Health coordinator that there were only 23 accredited psychiatrists and 124 psychologists available to meet the needs of the nation's population of 11 million in 2020 (Obert, 2021).

In addition to the broader deficiencies of the Haitian healthcare system, the healthcare institutions themselves and the practitioners that they employ face an entirely different subset of adversity. Considering Haiti remains the poorest country in the Latin America and Caribbean region (Overview, n.d.), Haitian healthcare providers are tasked with offering quality patient care without adequate facilities, resources or funding amidst an ongoing national crisis. Among the most notable sources of this resource-based adversity are the lingering financial consequences of the abrupt spike in fuel prices in 2018 (Doctors Without Borders, 2020). The 2018 crisis left healthcare facilities nationwide "unable to provide basic services," such as blood, drugs, oxygen or adequate staffing. Given the high demand for healthcare, in addition to the severe lack of resources and training to account for this demand, healthcare providers are subjected to an abundance of stressors that heighten their risk for burnout, anxiety and depression. After all, the WHO (2022) highlights excessive workloads in addition to unsafe or poor working conditions, understaffing, poor investment in career development and limited support from colleagues as risk factors for work-driven mental illness. Furthermore, extreme workloads are known contributors to Burnout Syndrome, "characterized by

fatigue, cynicism and reduced efficacy," which has been recently recognized in the 11th Revision of the International Classification of Diseases in 2019 (WHO, 2019a; Cotel et al., 2021; Nishimura et al., 2021). Failing to moderate the risk of burnout syndrome, anxiety and depression not only threatens the mental health of healthcare providers but also jeopardizes their ability to deliver high-quality patient care for general health services. Therefore, it is critical that Haitian healthcare providers are supported and provided with adequate resources so that their mental health can be prioritized in their practice and personal lives.

The present study aims to emphasize the acute need for a strategic mental health plan for healthcare professionals in Haiti by producing quantitative data that highlight the severity of anxiety and depression symptoms in this community. Although the data revealed in this study do not intend to estimate the prevalence of anxiety and depression disorders, the data aim to highlight the presence and severity of the symptoms consistent with these conditions in our sample of healthcare professionals. As such, we hypothesize that the majority of the healthcare providers in our sample may report anxiety and depression symptoms with intensities above the clinical threshold that recommends further psychological evaluation. In consideration of Haiti's ongoing state of crisis and the ripple effects of such a crisis on the healthcare system, we also predict that the providers in our sample may also report community violence and instability, as well as poor work environments as the factors that have the most prominent negative effect on their mental health. We intend to use the data provided through our research to inspire further investigation into how anxiety and depressive mental health conditions hinder the well-being and professional effectiveness of Haitian healthcare providers. With further investigation and advocacy for the development of a holistic and accessible trauma-informed care model, we hope that healthcare providers will obtain the mental health support they need to practice at their highest capacity.

## Method

### Research design

The present study uses an online cross-sectional design. During the data collection process, two primary dependent variables were measured to gain insight into the prevalence of symptoms associated with common mental health conditions in the sample of healthcare providers. The first primary measured variable was depression symptom severity, which was measured using PHQ-9. The second primary measured variable was anxiety symptom severity, which was measured using the GAD-7 questionnaire. Additional data were obtained via an original questionnaire that aimed to gauge mental health awareness, patient care impacts and mental health resource accessibility among Haitian healthcare providers and students.

### Participants

A sample of 106 Haitian adults (71 female and 35 male) from the western, southern, central, northern, north-west, Artibonite, Nippes and Grand'Anse regions of Haiti was recruited via convenience sampling. Twenty-six percent of the participants were aged 18–24 years ($N = 28$), 65% were aged 25–34 years ($N = 69$) and 9% were aged 35–44 years ($N = 9$). The sample was made up of both healthcare professionals and student interns, including physicians (55%), medical students (31%), psychologists (5%), nurses (4%),

nursing students (2%), midwives (2%) and students in other healthcare fields (1%). Of these healthcare professionals and students, 63% work in a hospital or clinical setting, 30% work or study in a university setting, 5% work in a non-profit setting, 1% work in a health center and 1% are currently unemployed. All of the participants met the following inclusion criterion: at least 18 years old, have work experience in one or more of the following fields – medicine, pharmacy, dentistry, nursing, laboratory, psychology, social work or medical education – or are a medical resident or intern student completing an accredited medical program or a program of study in another health field, and practice or study in Haiti. None of the participants were compensated in any way for their participation in this study.

### Materials

A self-administered online questionnaire developed using Google Forms software was used to collect data for this study. The form included five sections that each displayed a separate questionnaire. The first section included the informed consent form, which described the study in full and included necessary contact information for the participants. The second section included a basic demographic questionnaire. The third and fourth sections included a French-validated version of the GAD-7 and PHQ-9 questionnaires. The final section of the form presented an original survey developed by this study's leaders.

### GAD-7 questionnaire

The first section of the form used to collect data for this study included the official GAD-7 questionnaire developed by Spitzer et al. (2006). The GAD-7 presents a variety of general anxiety disorder-related symptoms and solicits patients to respond with a number that indicates the degree to which they have been bothered by those symptoms (e.g., feeling nervous, anxious or on edge, trouble relaxing and becoming easily annoyed or irritable). The questionnaire uses a 4-point Likert-type scale response format with possible responses ranging from 1 (not at all) to 4 (nearly every day). The numeric sum of the responses recorded for each question provides a cumulative score that indicates the severity of the patient's symptoms. Cumulative scores that fall within the scope of 0–4 correspond to minimal or no anxiety, scores between 5 and 9 correspond to mild anxiety, scores between 10 and 14 correspond to moderate anxiety and scores above 15 correspond to severe anxiety (Spitzer et al., 2006). Further psychological evaluation is generally recommended for patients who record cumulative scores ≥10 (Spitzer et al., 2006).

### Patient Health Questionnaire

The second section of the survey used to produce this study's data included PHQ-9, which was also developed by Spitzer et al. (1999). Like the GAD-7 questionnaire, the PHQ-9 solicits patients to respond with a number that indicates the degree to which they have been bothered by a variety of symptoms (e.g., little interest or pleasure in doing things, feeling tired or having little energy, and poor appetite or overeating) (Spitzer et al., 1999). The symptoms referenced in PHQ-9 have been associated in the fourth edition of the DSM with major depressive episodes (Marvin, 2011). The questionnaire uses a 4-point Likert-type scale response format with possible responses ranging from 1 (not at all) to 4 (nearly every day). Cumulative scores that fall within the range of 0–4 correspond to no depression or normal levels of depression, scores between 5 and 9 correspond to mild depression, scores between 10 and

14 correspond to moderate depression, scores between 15 and 19 correspond to moderately severe depression and scores above 20 correspond to severe depression (Spitzer et al., 1999).

### Original questionnaire

The final section of the questionnaire included an original survey designed for this study. The questionnaire used a forced choice response format to gain insight into the participant's outlook on mental health and how it impacts their healthcare practice. The full questionnaire is shown in Table 1.

### Procedure

The complete questionnaire was made available on several online outlets from March 11, 2024, to May 19, 2024. During this period, the questionnaire was distributed via Google Groups and WhatsApp to a variety of Haitian healthcare provider groups and networks with the support of the Social Medicine Alumni Haiti Association, a network of healthcare professionals working for

**Table 1.** The contents of the original French survey, which was developed by Youri Encelotti Louis, Brooke Betson and Samy Auguste

| |
|---|
| **Have you ever been treated for a mental health condition?**<br>( ) Yes<br>( ) No<br>( ) I am not sure |
| **Which of the following factors have a negative impact on your mental health?**<br>( ) Work environment<br>( ) Romantic relationships<br>( ) Relationships with family members<br>( ) Violence in the community<br>( ) Your socioeconomic status<br>( ) Political instability<br>( ) National socioeconomic status<br>( ) Other healthcare problems<br>( ) None of these have a negative impact on my mental health |
| **Which of the following categories has the most negative impact on your mental health?**<br>( ) Personal problems (romantic or family relationships, personal socioeconomic status)<br>( ) Problems in your community (violence, political instability, insecurity)<br>( ) Work environment<br>( ) None of these |
| **Do you agree that your mental health has a negative impact on your work?**<br>( ) Yes<br>( ) No<br>( ) I am not sure |
| **Do you agree that your mental health is preventing you from consistently providing quality patient care?**<br>( ) Yes<br>( ) No<br>( ) I am not sure |
| **Do you have a mental support system at your workplace or school?**<br>( ) Yes<br>( ) No<br>( ) I am not sure |
| **If you answered yes to the question above, do you agree that this mental health support system is effective and reliable?**<br>( ) Yes<br>( ) No<br>( ) I am not sure |

health equity and social justice in Haiti. Notice of confidentiality was included in the questionnaire, and informed consent was obtained from all participants. Upon providing consent and completion of the GAD-7, PHQ-9 and original questionnaire, the answers submitted by each participant were analyzed and subjected to statistical analysis using the Mini Tab and Microsoft Excel software. In addition to the identification of the mean scores produced by the male and female participants, one-way analysis of variance (ANOVA) tests were completed to compare both the mean GAD-7 and PHQ-9 scores between sexes and each age category to identify any significant differences. The Pearson correlation coefficient was also calculated to identify any significant relationship between the GAD-7 and PHQ-9 scores that might suggest anxiety and depression symptom comorbidity in our sample of participants. Upon completion of the questionnaire, a comprehensive list of mental health resources, including both public and private psychology clinics, was automatically populated for all participants. Furthermore, these resources were disseminated to all participants whose GAD-7 and PHQ-9 scores surpassed the threshold necessitating further evaluation.

### GAD-7 results

The mean GAD-7 score among all participants was 8.64 (SD = 5.82), which falls within the mild anxiety category. More specifically, 31% of the sample produced a GAD-7 score corresponding to minimal anxiety symptom severity, 28% produced a score corresponding to mild symptom severity, 16% produced a score corresponding to moderate symptom severity and 25% produced a score corresponding to severe symptom severity. Therefore, the overall prevalence of anxiety symptoms in our sample was 69%, and 41% of participants produced GAD-7 scores at or above the clinical threshold that recommends further psychological evaluation. The distribution of the GAD-7 symptom severity scores is illustrated in Figure 1. Furthermore, a one-way ANOVA test found a significant difference between female GAD-7 scores ($M$ = 9.56, SD = 6.07) and male GAD-7 scores ($M$ = 6.77, SD = 4.82), $F$ $(1,104)$ = 5.64, $p$ = 0.019. Additionally, there was also a significant difference between the 18 and 24 age category ($M$ = 10.44, SD = 5.29), the 25 and 34 category ($M$ = 8.42, SD = 6.02) and the 34 and 44 age category ($M$ = 5.3, SD = 3.71), $F(2,103)$ = 3.12, $p$ = 0.049. This test suggests that the anxiety symptom severity of the participants in our sample was most severe in the 18–24 age category and decreased in the subsequent categories.

### PHQ-9 results

The mean PHQ-9 score among all participants was 11.2 (SD = 5.93), which falls within the category of moderate depression. More specifically, 16% of the sample produced a PHQ-9 score corresponding to minimal depression symptom severity, 26% produced a score corresponding to mild symptom severity, 29% produced a score corresponding to moderate symptom severity, 18% produced a score corresponding to moderately severe symptom severity and 11% produced a score corresponding to severe depression symptom severity. Therefore, a combined 58% of our sample produced scores at or above the clinical threshold that recommends further psychological evaluation and the overall prevalence of depressive symptoms among this sample is 84%. The distribution of the PHQ-9 score categories is illustrated in Figure 2. Furthermore, a one-way ANOVA test found a significant difference between female PHQ-9 scores

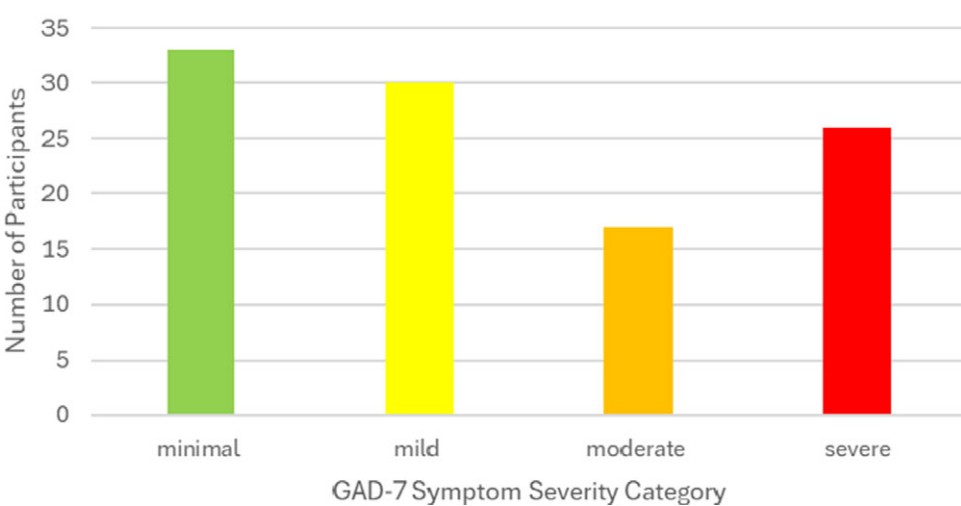

**Figure 1.** The distribution of the GAD-7 anxiety symptom score categories in our sample (*N* = 106).

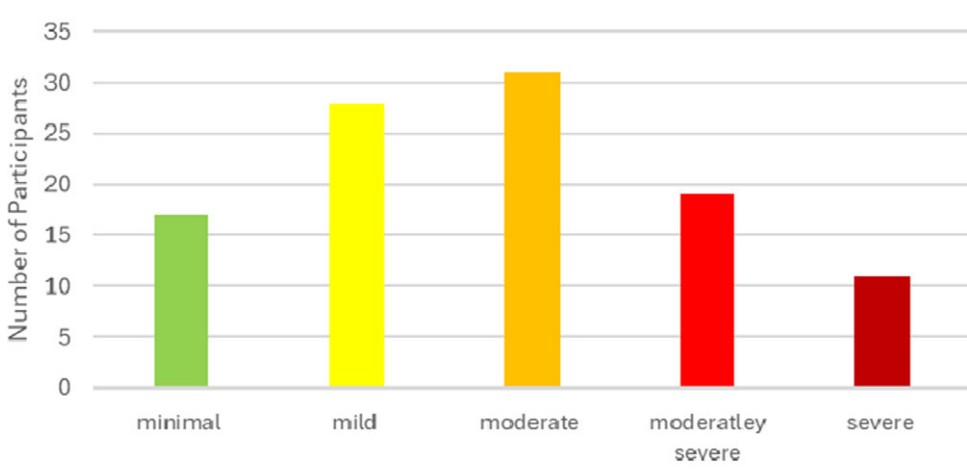

**Figure 2.** The distribution of the PHQ-9 depression symptom severity scores in our sample (*N* = 106).

($M$ = 12.47, SD = 6.11) and male PHQ-9 scores ($M$ = 8.66, SD = 4.69), $F(1,104)$ = 10.52, $p$ = 0.002. This test suggests that the female healthcare providers in this sample reported more severe depressive symptoms compared with the male participants. Additionally, there was also a significant difference between the 18–24 age category ($M$ = 13.52, SD = 6.61), the 25–34 category ($M$ = 10.97, SD = 5.54) and the 34–44 age category ($M$ = 7.3, SD = 4.37), $F(2,103)$ = 4.62, $p$ = 0.012.

### Original questionnaire

Ninety percent of the participants reported having received no previous mental health treatment, 6% reported having previous treatment and 4% reported being unsure about their previous experience with mental health treatment. When asked to identify the factor that has the most substantial negative effect on their mental health, 11% of the participants responded with their work environment, 74% responded with problems within their community (e.g., violence, social/economic insecurity and political instability), 16% responded with problems in their personal lives and 6% responded with their work environment. When asked whether they agree that the state of their mental health negatively impacts their work, 54% agreed, 24% disagreed and 22% were unsure. Similarly, when asked if they agree that their mental health is preventing them from consistently providing high-quality patient care, 31% agreed, 39% disagreed and 30% were unsure. When asked if they have a mental health support system in their workplace or school, 16% agreed, 76% disagreed and 8% were unsure. Among the participants who agreed to the aforementioned question, 50% agreed, 24% disagreed and 26% were unsure if their support system was effective. Furthermore, PHQ-9 and GAD-7 scores were found to be positively correlated, $r(104)$ = 0.767, $p$ < .01. This indicates the positive relationship between depressive and generalized anxiety symptoms in the

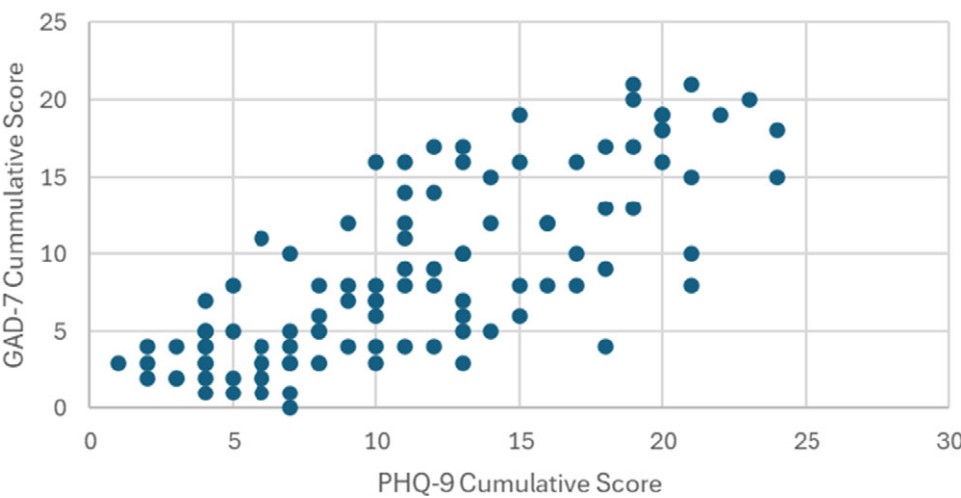

**Figure 3.** The positive relationship between the PHQ-9 and GAD-7 cumulative scores in our sample (*N* = 106).

sample of Haitian healthcare providers. This relationship is illustrated in Figure 3.

## Discussion

The data produced in this present study partially support our hypothesis that our participants would report both anxiety and depression symptom severities beyond the threshold for further recommended psychological evaluation. Regarding our measurement of anxiety symptom severity, the data found did not support our hypothesis as only 41% of the participants reported GAD-7 scores at or above the threshold identified by Spitzer et al. (2006). In contrast, the depression symptom severity reported by our participants was consistent with our hypothesis, as the majority of our participants (58%) reported PHQ-9 scores that were at or above the threshold identified by Spitzer et al. (1999). It is important to emphasize that 69% and 84% of participants were experiencing respectively mild, moderate and severe symptoms of anxiety and depression. According to our data, females and the younger age groups were the most affected by these symptoms. Although the contribution of these factors was not measured directly in this study, our prediction that community violence and insecurity, in addition to the work environment, would be reported as the most influential on the mental health of our sample of healthcare providers was proven to be partially correct. More specifically, the majority of our sample (74%) reported "problems in the community (e.g., violence, insecurity and national socioeconomic status)" as the factor that has the most significant negative effect on their mental health. However, only 11% of the sample identified "work environment" as the factor with the most significant negative effect on their mental health. Considering the data produced from this study partially supports our hypothesis and demonstrates the prevalence of mental illness symptoms across a wide spectrum of severity in our sample of healthcare providers, we believe that the data demonstrate the urgent need for an adequate mental health support system to be put in place for healthcare professionals in Haiti while we also acknowledge the need to address the root cause of the inequalities.

In addition to the dismaying context of the current humanitarian crisis and the detrimental effect that it has on public health,

Haitian health professionals face an abundance of occupational stressors in their respective workplaces that expose them to further risk. Despite these conditions, an astounding majority of participants in this study reported having no access to mental health resources or support systems. In consideration of the international recognition of Haiti's poverty crisis, these findings align with the data produced by Gureje and Ayinde (2022), which highlight the significant treatment gap for mental health disorders in resource-constrained and LMICs. Furthermore, this mismatch between the expectations for healthcare providers and the lack of resources made available to support them increases the susceptibility of health providers to various stress-related mental health complications in consideration of the WHO's (2022) identification of excessive workloads and workplaces as primary risk factors for occupational-driven mental illness. Ultimately, our results suggest health professionals in Haiti do not have access to the necessary mental health resources when needed. As a result, these professionals, who often bear the responsibility of providing life-saving care, are potentially practicing in a state of stress and are being predisposed to mental illness or burnout.

The data produced in this study also suggest broader implications that extend beyond the community of Haitian healthcare professionals. After all, a concerningly substantial portion of participants reported either an unsure or concrete belief that their mental health negatively affects their work. In another sense, our data suggest that the lack of supportive mental health resources in healthcare workplaces not only adversely affects the well-being of the providers themselves but may also adversely affect their capacity to enhance the well-being of their patients and provide high-quality general health services. Considering multiple studies, such as that of Grover et al. (2018), it is suggested that there is a positive correlation between stress and depressive symptoms in healthcare providers and negative patient outcomes; continuing to knowingly ignore the lack of mental health support within the healthcare provider community may inevitably undermine the effectiveness of the healthcare system in its entirety. Furthermore, previous research (e.g., Wainberg et al., 2017) suggests that systematic factors, including resource inaccessibility and stigmatization, are among the most impactful hindrances on mental health, particularly in LMICs. Thus, to properly support a nation that is engulfed

in near-constant instability, all efforts should be exhausted to address the upstream systematic barriers and oppressive forces that are hindering access to the necessary supportive resources, medical supplies and health infrastructures that will allow Haitian healthcare providers to provide the highest degree of care that is possible.

The complexity of mental health perception in Haiti must also be considered in the pursuit of enhanced mental health awareness and resource accessibility in healthcare communities and beyond (Khoury et al., 2012). Considering Haiti's roots in spirituality and Vodou (Cain, 2014), mental illness perception tends to be more cosmocentric than anthropocentric for many Haitians, which often causes them to de-emphasize symptoms of mental illness from a clinical standpoint and to seek spiritual support (Auguste & Rasmussen, 2019). This reality is coupled with the lack of clinical organizations that offer mental health services nationwide, with even less accessibility in rural areas of Haiti. To improve access to mental healthcare services in Haiti, cultural competency must be at the center of the interventions in conjunction with a community-based approach capable of generating a partnership between cultural and clinical worldviews (Cross, 1989; Colin n.d.). In doing so, mental health awareness and resources can be prioritized not only for the general public but also for the valued healthcare providers who generously offer their services.

Although this research provides useful insights into the prevalence of depressive and anxiety symptoms among Haitian healthcare providers and the effects of those symptoms on general patient care, the study was impacted by limitations. Our sample size ($N = 106$) is relatively small compared with the full population of Haitian healthcare providers and students across all domains. Additionally, this study's mechanism of sampling likely adversely affects the generalizability of the results. Due to the geographic barrier that separates us from the community in which our target population resides, we had to collect data remotely through convenience sampling. More specifically, the questionnaire was distributed predominantly to networks and healthcare groups that were known to this study's authors, which limited our pool of potential participants. The remote nature of the data collection process also led us to implement a forced-choice response format. Although this format was effective in simplifying the data collection process and enhancing the questionnaire's ease of use for the participants, it is possible that participant responses were constrained and that the available options did not allow our participants to express the full extent of their beliefs. The general lack of international validation of the GAD-7 screening tool is another limitation of this study that may have implications for the relative significance of its results. We have used the validated French version of the questionnaires from The Centre for Addiction and Mental Health in Canada for consistency, as French is the primary educational language of the participants and we could not find any validity data on a Creole version of the GAD-7 questionnaire. Although the findings of Marc et al. (2014) suggest strong evidence of psychometric adequacy, effectiveness and reliability of the Haitian Creole translation of the PHQ-9 screening measure in Haiti, the GAD-7 screening tool has not yet obtained Haitian validation despite its accreditation in Lesotho, Mexico, and the United States (Partners in Health, 2021). Therefore, until official validation of the GAD-7 screening tool is achieved in Haiti, it cannot be confirmed that this study's results accurately reflect the prevalence and severity of anxiety symptoms in our sample of healthcare providers.

Despite its limitations, we believe that several critical opportunities can arise from this study. Among these opportunities is the generation of awareness and desire to advance the importance of mental health resource accessibility for Haitian healthcare providers. To reinforce the findings of this study and to advance this message, future research aimed at validating the GAD-7 and PHQ-9 screening tools in Haiti would be of tremendous benefit. Future research in this area of study might also benefit from a more representative sampling method and a larger sample of participants. Upon completion of additional valid research that reinforces our findings, appropriate strides can be taken, to ensure that Haitian healthcare does not remain unidirectional. After all, the literature referenced throughout this article suggests that the well-being of a provider has an instrumental impact on their ability to care for their patients (e.g., Ryu and Shim, 2021; Wright et al., 2022). Therefore, we hope that the results of this study and future research inspire change and the desire to develop an adequate support system that accounts for the comprehensive health of Haitian healthcare providers so that they can deliver high-quality patient care.

## Conclusion

Providing readily accessible and effective mental healthcare services remains a significant challenge in Haiti. Despite the known susceptibility to mental illness and other health conditions, there is much to be done before the existing treatment gap for mental health disorders can be filled. The lack of accessibility and strategic plan for mental healthcare services in Haiti constitutes another challenge for the broken Haitian healthcare system and also for Haitian healthcare providers who carry the tremendous responsibility of providing quality patient care amidst an ongoing humanitarian crisis and in resource-constrained settings. Social determinants of health, such as socioeconomic status and environment, can seriously affect mental health. It was demonstrated in the data that the most detrimental factor that impacted participants' mental health was their living environment. The results of this study show the importance of further investigation into the burden of mental health disorders in the health sector in Haiti. Based on GAD-7 and PHQ-9 questionnaire results, 84% of participants had mild to severe symptoms of depression and 69% had mild to severe symptoms of anxiety. Despite these findings, a surprising 76% of participants reported not having an adequate mental health support system. Further investigations that encompass larger and more representative sample sizes are necessary for developing an informed overview of the prevalence of depression and anxiety disorders among healthcare professionals in Haiti. However, the data produced by this study serve as an urgent call to action to improve access to mental healthcare services in clinical settings and community-based institutions at a local level.

**Open peer review.** To view the open peer review materials for this article, please visit http://doi.org/10.1017/gmh.2025.7.

**Data availability statement.** Data available within the article or its supplementary material.

**Acknowledgements.** We thank the EqualHealth and Social Medicine Alumni Haiti organizations for their support in the questionnaire distribution process and continuous support throughout this project. Special thanks to Dr. Amy Colleen Finnegan, Sociologist, Dr. Anna Johnson, Psychologist, and Dr. Cidna Valentin, Clinical Psychologist, for their contributions to this paper.

**Author contribution.** All the listed authors meet the criteria for authorship according to the Cambridge University Publishing Ethics Guidelines. Y.E.L. and B.B. conceived the study and are the principal authors of the manuscript. S.A. supported data collection. Y.E.L. provided with the design of methodology

and managed responsibilities. B.B. analyzed the data under the supervision of Y.E.L. All authors contributed to the data validation, edited the paper and approved it for submission.

**Financial support.** This research received no specific grant from any funding agency in the public, commercial or not-for-profit sectors.

**Competing interest.** The authors declare no competing interests.

**Ethical standard.** This study was conducted in accordance with relevant international ethical and legal standards for research. This study was reviewed and approved by the Institutional Review Board (IRB) at the University of St. Thomas, Saint Paul, MN (2145475-2). This study was also reviewed and approved by the IRB at La Paix University Hospital, Port-au-Prince, Haiti.

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
