## [Reviewer Report]

The manuscript discusses important discoveries about the mental health of Haitian healthcare providers. It highlights the prevalence of depression and anxiety among them and emphasizes the impact of social determinants such as community violence, economic and social instability, and poverty. The finding that 76% of participants reported either no awareness or no access to mental health support underscores the urgent need for targeted interventions. While the study is well-conducted and adds valuable insights to the existing literature, one recommendation for improvement is to explain the translation process used for the questionnaire. Since Haitian Creole is unique, it is crucial to preserve the questions' content and meaning during translation. Including a brief overview of the translation and back-translation process in the methods section would demonstrate the efforts made to maintain the accuracy and cultural relevance of the survey instrument. This addition would enhance the study’s rigor and ensure the validity of the findings across linguistic contexts.

---

## [Reviewer Report]

I would like to make few comments after my third review of the paper:

1) Beside the PHQ-9 and the GAD-7 as measurement tools, has the Original Questionnaire used in the research gone through a validation process ? If yes, this should have been mentioned.

2) 1% of the participants is unemployed. Could more precision have been brought about how long they have been out of the Health System as providers. If their unemployment status has been for long, why should they be part of the sample ?

3) 39% of the participants have declared that their MH situation, although impacted negatively is not preventing them from providing high quality patient-care in one hand. In another hand 54% their MH is impacted negatively. The accurate data would be 16%. Would it be a matter of “professional desirability” ? I would more likely to believe that is higher than 16 % of Staff with impacted MH, making the paper even more pertinent.

General Comments:

As a relentless advocate for staff Wellness at the international level, I strongly believe that this paper can be appreciated as a great effort to raise awareness about the MH needs of the Health Care providers in Haiti and a call towards the decision makers from both private and public sectors to take action addressing this issue in a context of social violence and of political instability that is not ending tomorrow.

This is a great contribution to a noble cause.

---

## [Editor Report]

Sample is comprised of practicing health professionals and students. Could you clarify if students also provide direct services to the population in the context of Haiti?

Could you provide some of the available validation and psychometric information on the PHQ-9 in the context of Haiti when describing the measures? (i.e., reliability, transition to kreyòl, etc.). The GAD-7 although not validated, did you translated the measure? If so, please mention this when describing the measure.

In your discussion, you mention the limitations of the data collection format and the use of measures yet to be validated. Despite these limitations that hamper the accurate representation of local idioms of distress, the study found high frecuency of symptoms among the sample. This is an important finding and it would strengthen this manuscript if the authors could provide some recommendations to policy makers, organizations, and providers as to how they can potentially support the available working force.

---

## [Reviewer Report]

The manuscript under review presents a crucial study on the mental health of Haitian healthcare providers amidst the ongoing humanitarian crisis and systemic healthcare limitations in Haiti. The findings emphasize the urgent need for mental health resources in resource-limited settings, highlighting the severe economic, social, and environmental stressors contributing to a widespread mental health crisis.

The study employs reliable tools like the GAD-7 and PHQ-9 to assess anxiety and depression, enhancing its reliability and comparability to other research. By focusing on social determinants, it provides a comprehensive view of the mental health landscape for Haitian healthcare workers.

The results indicate high levels of depression and anxiety, coupled with low access to mental health support, highlighting the urgent need for advocacy and resource allocation. The manuscript strongly advocates for immediate intervention to support healthcare workers, emphasizing the need for additional resources and policy changes.

Recommendation: I recommend accepting this manuscript for its valuable insights and evidence-based call to action addressing mental health needs in Haiti. It significantly contributes to the literature on global health disparities and paves the way for impactful policy and practice changes. However, I do have some Questions-

1. Have there been any observed increases in mental health episodes among young Haitians?

2. Are there growing numbers of cases of suicidal ideation in this demographic?

3. Do clinics report a rise in the number of young Haitians presenting with mental health concerns?

---

## [Reviewer Report]

This paper is putting out a veil that was hiding the mental health challenges faced by the staff and their increasing need Of wellness. Of course it did not provide a clear pathway of the actions not be taken, however it is a good start. There are lessons to learn from the implementation champions like PIH and other NGOs. This can be very useful and helpful.